# Influence of elevated temperature on the nutritional profile of Chickpea (*Cicer arietinum* L.) Seeds

Uday Chand Jha[1,2]*, Marilyn Warburton[3], Harsh Nayyar[4], Sadiah Shafi[2],
Ignacio A. Ciampitti[5,6], Ashis Ranjan Udgata[1], Kadambot H. M. Siddique[7],
P. V. Vara Prasad[2,5]*

1 Indian Council for Agricultural Research (ICAR) - Indian Institute of Pulses Research (IIPR), Kanpur, Uttar Pradesh, India, 2 Feed the Future Innovation Lab for Collaborative Research on Sustainable Intensification, Kansas State University, Manhattan, Kansas, United States of America, 3 USDA ARS, Western Regional Plant Introduction Station, Pullman, Washington, United States of America, 4 Department of Botany, Panjab University, Chandigarh, India, 5 Department of Agronomy, Kansas State University, Manhattan, Kansas, United States of America, 6 Department of Agronomy, Purdue University, West Lafayette, Indiana, United States of America, 7 The University of Western Australia – Institute of Agriculture, The University of Western Australia, Crawley, Perth, Australia

* u9811981@gmail.com (UCJ); vara@ksu.edu (PVVP)

## Abstract

Increasing occurrences of episodic heat stress significantly affect crop quality traits, including those of chickpea (*Cicer arietinum* L.). The adverse effectof heat stress on seed quality was evaluated by cultivating eight chickpea genotypes under non-stress and heat stress conditions, with temperatures set at 25/15°C and 35/20°C, respectively. The genotypes exhibited notable genetic variations in *"seed carbon (C, %), protein (%), phosphorus (P, %), potassium (K, %), magnesium (Mg, %), sulfur (S, %),* and *manganese (Mn, ppm)"* concentrations under both conditions. However, no significant variations were observed for seed (S%), seed iron (Fe, ppm), and zinc (Zn, ppm), concentrations under NS conditions or seed copper (Cu, ppm) under heat stress conditions. The genotype (G) × temperature (T) interaction was significant for all traits except for seed K. Correlation analysis revealed positive associations between seed C and protein, seed Mg and P, and seed protein and S under non-stress (NS) conditions. Under heat stress, significant correlations were observed between seed protein and Mg, and seed protein and P. In contrast, significant negative correlations were observed between seed Ca and K under NS conditions and seed Ca and K and seed Fe and Cu under heat stress conditions. The adverse effects of heat stress on nutritional quality and seed yield underscore the necessity for continued research into developing heat-tolerant chickpea cultivars with enhanced seed nutritional traits.

**Data availability statement:** All relevant data are within the paper and its Supporting Information files.

**Funding:** This study was financially supported by the United States Agency for International Development (USAID) in the form of a Cooperative Agreement award (AID-OAA-L-14-00006), received by PVVP, as part of Feed the Future Innovation Lab for Collaborative Research on Sustainable Intensification (SIIL) at Kansas State University, including contribution number 25-185-J from the Kansas Agricultural Experiment Station. No additional external funding was received for this study.

**Competing interests:** The authors have declared that no competing interests exist.

## Introduction

Chickpea (*Cicer arietinum* L.) is a nutritionally rich pulse crop that serves as a vital source of bioavailable proteins, macronutrients, micronutrients, and vitamins, helping to alleviate malnutrition and 'hidden hunger,' particularly among low-income populations [1]. Chickpea contributes significantly to food security, with an annual global production of approximately 18.1 million tons (Mt) harvested from 14.8 million hectares (Mha) [2]. India is the leading producer, accounting for 11.08 Mt from 9.7 Mha annually [2]. As a cool-season legume, chickpea is well adapted to mild-temperature environments but highly susceptible to heat stress during vegetative and reproductive stages [3]. In northern and southern Indian regions, the increasing frequency of heat waves (>32°C) during the growing season, especially during the reproductive stage, has led to substantial yield losses [3–6] and has been attributed to climate change [7–9]. High-temperature stress at anthesis adversely affects pollen germination and viability, stigma receptivity, fertilization, and pod and seed development, ultimately reducing yield [3,6,10,11]. One study showed that exposure to high temperatures (30–35°C) during flowering reduces chickpea yield significantly [12]. Other studies reported thattemperatures exceeding 35°C reduced chickpea pod and seed set, decreasing yield by up to 39% [13,14]. Moreover, each additional degree above the optimal threshold (32°C) decreased chickpea yield by 15% [15] and increasing crop growing temperature by 4°C above ambient conditions reduced chickpea yield by 9–41% [16]. Beyond yield losses, heat stress also affects seed nutrient composition [17–19]. Chickpea seeds are rich in essential nutrients such as carbon (C), protein, phosphorus (P), magnesium (Mg), potassium (K), and sulfur (S), and micronutrients like iron (Fe), zinc (Zn), manganese (Mn), and copper (Cu).While extensive research has focused on the impact of heat stress on cereal grain quality [20], there is comparatively little informationon its effects on seed nutritional traits in grain legumes [19], including chickpea. In lentil, heat stress has been linked to an 8% reduction in starch content [21] and a decrease in seed protein concentration [22]. Similarly, heat stress decreased Zn, Fe, Ca, and Mg concentrations in lentil compared to non-stress conditions [23]. In chickpea, limited studies suggest a significant reduction in Fe and Zn concentrations under heat stress [24], with heat-sensitive genotypes showing Fe reductions of up to 59% and Ca reductions of 54% [18]. Given these findings, further investigation is necessary to understand how heat stress affects chickpea seed nutritional traits. This study aims to bridge this knowledge gap by evaluating the influence of elevated temperature on chickpea seed nutrient composition. The results will provide critical insights into developing heat-resilient chickpea varieties with improved nutritional quality.

## Materials and methods

### Plant materials

Eight diverse chickpea genotypes—PI 372596, PI 360688, PI 368485, PI 598080, PI 513144, PI 360691, PI 518255, and Gokce (see S1Table in S1 File)—were evaluated for seed yield and nutrient quality responses to heat stress. Seeds were sourced from

the USDA ARS, Western Regional Plant Introduction Station, Pullman, Washington, USA. The experiment was conducted under NS conditions at Kansas State University, Manhattan, in 2024.

## Experimental set up

The study followed a randomized complete block design (RCBD) with three biological replicates, each comprising a single pot containing two plants. Seeds were sown on 29.1.2024 in 20 cm diameter pots filled with potting soil constituting (peat moss, vermiculite, compost, coir, garden soil, sand, manure, vermicompost and limestone) (Fafard®3B Mix/Metro-Mix®830, SUNGRO Horticulture, Agawam, MA, USA) "and maintained in a greenhouse for 60 days until flower initiation. At this stage, plants were transferred to a growth chamber last week of March, 28.3.2024) for controlled temperature treatments under non-stress conditions (NS; 25/15°C day/night) or heat stress conditions (HS; 35/20°C day/night) (see Fig.1). Plants were exposed to photosynthetically active radiation (400–700 nm) of 600 µmol m$^{-2}$ s$^{-1}$ under a 12 h photoperiod provided by cool fluorescent lamps. The relative humidity was maintained at 60%, and plants were watered regularly to prevent drought stress. A consistent nutrient supply was maintained (for details, see Jha et al. [25]). Temperature data were recorded using a HOBO®data logger (Onset Computer Corporation, Bourne, Massachusetts, USA)"(S1 Fig in S1 File). Plants were harvested at physiological maturity for seed yield and nutritional quality analysis.

## Analysis of seed nutritional components

Seed samples (0.5 g per genotype) were collected in triplicate and analyzed for various nutritional components: protein (%), C (%), P (%), K (%), Mg (%), Ca (%), S (%), Mn (ppm), Cu (ppm), Fe (ppm), and Zn (ppm). All analyses were performed at the Kansas State Soil Testing Laboratory (Manhattan, KS, USA). Total carbon (C) and nitrogen (N) concentrations were determined using a LECO TruSpec CN Carbon/Nitrogen combustion analyzer (LECO Corporation, St. Louis, MO, USA). Concentrations of P, K, Mg, Ca, S, Mn, Cu, Fe, and Zn were measured using a nitric-perchloric digest method,

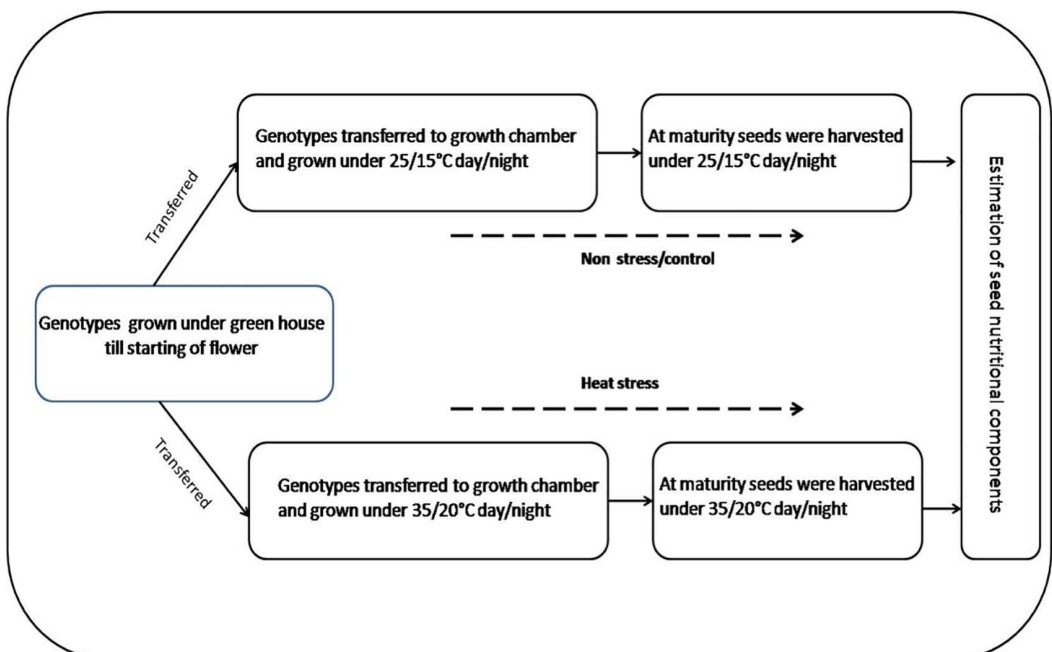

**Fig 1. Graphical representation of methodology used in the experiment.**

followed by analysis with an Inductively Coupled Plasma (ICP) spectrometer (Model 5800 ICP OES, Agilent Technologies, Santa Clara, California, USA) [26,25]. Seed protein concentration was calculated using the Kjeldahl method, multiplying N concentration by 6.25 [27]. The entire nutrient estimation process followed the methodology outlined by Jha et al. [25].

### Seed yield per plant (SYP)

At maturity, plants were harvested, and seed yield per plant (SYP) was determined by threshing the seeds from two plants per replicate and calculating the average yield.

### Data analyses

Analysis of variance (ANOVA) was conducted using OPSTAT software, with the least significant difference (LSD) calculated at the 5% and 1% significance levels. Correlation analyses for seed nutrient traits and seed yield per plant (SYP), principal component analysis (PCA), and genotype clustering were all performed using RStudio.

## Results

Under NS, ANOVA revealed significant genetic variation among the eight chickpea genotypes for all nutrient parameters except seed iron (Fe), zinc (Zn), and copper (Cu) concentrations. The observed ranges under NS condition were: protein (14.2–22.9%), carbon (C) (40.4–41.2%), phosphorus (P) (0.38–0.55%), potassium (K) (1.21–1.58%), calcium (Ca) (0.07–0.17%), magnesium (Mg) (0.15–0.19%), sulfur (S) (0.21–0.25%), manganese (Mn) (25.2–45.7 ppm), zinc (Zn) (67.7–83.6 ppm), copper (Cu) (2.13–4.96 ppm), iron (Fe) (58.8–101.2 ppm), and seed yield per plant (SYP) (4.1–8.77 g) (Tables 1 and 2). Conversely, under high salinity (HS) conditions, significant genetic variation was observed for all traits except seed Fe concentration, with the following ranges: protein (16.3–26.4%), C (40.1–41.18%), P (0.44–0.65%), K (1.29–1.56%), Ca (0.09–0.16%), Mg (0.16–0.20%), S (0.24–0.30%), Mn (26.9–52.8 ppm), Zn (67.9–91.4 ppm), Cu (1.73–3.20 ppm), Fe (41.5–81.3 ppm), and SYP (2.6–4.9 g) (Tables 1 and 3). Furthermore, a combined ANOVA indicated significant genotype × treatment (G × T) interactions for all traits except seed K concentration (Table 4 and S2Table in S1 File).

### Protein and carbon concentration

The average protein concentration increased from 18.38% to 19.34% under HS (Fig 2), whereas the carbon concentration slightly decreased from 40.88% to 40.62%. The protein concentration exhibited significant genotype and G × T interaction

**Table 1. Genetic variability for chickpea seed nutrient components under non-stress (control) and heat stress conditions.**

|  | Treatment | Protein (%) | C (%) | P (%) | K (%) | Ca (%) | Mg (%) | S (%) | Mn (ppm) | Zn (ppm) | Cu (ppm) | Fe (ppm) | SYP (g) |
|---|---|---|---|---|---|---|---|---|---|---|---|---|---|
| Minimum | NS | 14.2 | 40.42 | 0.38 | 1.21 | 0.07 | 0.15 | 0.21 | 25.2 | 67.67 | 2.13 | 58.8 | 4.1 |
|  | HS | 16.3 | 40.12 | 0.44 | 1.29 | 0.09 | 0.16 | 0.24 | 26.9 | 67.9 | 1.73 | 41.53 | 2.6 |
| Maximum | NS | 22.9 | 41.22 | 0.55 | 1.58 | 0.17 | 0.19 | 0.25 | 45.7 | 83.63 | 4.96 | 101.2 | 8.77 |
|  | HS | 26.4 | 41.18 | 0.65 | 1.56 | 0.16 | 0.2 | 0.3 | 52.8 | 91.43 | 3.2 | 81.3 | 4.9 |
| Mean | NS | 18.38 | 40.88 | 0.48 | 1.36 | 0.121 | 0.170 | 0.23 | 32.98 | 74.93 | 2.86 | 74.5 | 6.76 |
|  | HS | 19.34 | 40.62 | 0.54 | 1.44 | 0.116 | 0.178 | 0.27 | 35.02 | 75.68 | 2.37 | 63.42 | 3.39 |
| Standard error | NS | 1.21 | 0.11 | 0.02 | 0.041 | 0.014 | 0.005 | 0.005 | 2.52 | 2.25 | 0.32 | 5.98 | 0.58 |
|  | HS | 1.28 | 0.14 | 0.021 | 0.04 | 0.010 | 0.005 | 0.008 | 3.1 | 3.12 | 0.17 | 4.66 | 0.26 |
| Coefficient of variation | NS | 18.60 | 0.76 | 11.6 | 8.64 | 33.1 | 8.3 | 6.3 | 21.5 | 8.5 | 31.7 | 22.7 | 24.3 |
|  | HS | 18.70 | 0.95 | 11.1 | 7.6 | 23.8 | 7.2 | 8.5 | 25.3 | 11.7 | 20.6 | 20.7 | 22.1 |

NS: non-stress; HS:heat stress
SYP: seed yield per plant

Table 2. **Seed nutritional and seed yield per plant (SYP) for eight chickpea genotypes under non-stress conditions.**

| Genotype | Protein (%) | Carbon (%) | Phosphorus (%) | Potassium (%) | Calcium (%) | Magnesium (%) | Sulfate (%) | Copper (ppm) | Iron (ppm) | Manganese (ppm) | Zinc (ppm) | SYP (g) |
|---|---|---|---|---|---|---|---|---|---|---|---|---|
| PI368485 | 22±0.56[a] | 41.223±0.07[a] | 0.501±0.01[ab] | 1.39±0.03[ab] | 0.066±0.002[c] | 0.162±0.002[b] | 0.226±0.003 | 2.367±0.75[b] | 58.833±2.5 | 26.5±0.8[c] | 70.267±3.2 | 7.57±0.23[b] |
| PI518255 | 17±1.8[bc] | 40.733±0.1[b] | 0.498±0.01[ab] | 1.30±0.04[b] | 0.162±0.003[a] | 0.165±0.002[b] | 0.228±0.004 | 2.567±0.15[b] | 65.333±1.58 | 45.667±1.1[a] | 76.133±3.5 | 8.7±0.18[a] |
| PI360691 | 21.967±1.4[a] | 41.11±0.13[a] | 0.505±0.01[ab] | 1.39±0.02[ab] | 0.102±0.01[b] | 0.192±0.01[a] | 0.253±0.01 | 3.2±0.6[b] | 80.833±22.7 | 26.433±2.6[c] | 69.633±1.2 | 4.1±0.15[d] |
| PI598080 | 22.9±1.36[a] | 40.69±0.12[b] | 0.483±0.01[b] | 1.36±0.07[b] | 0.151±0.02[a] | 0.169±0.01[b] | 0.25±0.02 | 2.767±0.03[b] | 101.167±2.57 | 38.3±1.8[ab] | 83.033±5.2 | 7.8±0.12[b] |
| PI360688 | 17.833±0.3[b] | 41.123±0.1[a] | 0.408±0.01[c] | 1.4±0.01[ab] | 0.088±0.01[bc] | 0.145±0.01[c] | 0.222±0.01 | 2.133±0.2[b] | 97.7±15.1 | 33b±3.4[c] | 78.867±2.7 | 5.16±0.01[c] |
| Gocke | 15.733±1.24[bc] | 41.16±0.2[a] | 0.382±0.04[c] | 1.22±0.16[b] | 0.153±0.02[a] | 0.159±0.01[bc] | 0.222±0.02 | 4.967±0.5[a] | 63.167±14.7 | 25.167±5.3[c] | 67.667±17.7 | 5.3±0.01[c] |
| PI513144 | 15.433±0.12[bc] | 40.42±0.05[c] | 0.547±0.01[a] | 1.57±0.01[a] | 0.077±0.01[bc] | 0.189±0.001[a] | 0.235±0.002 | 2.467±0.2[b] | 59.233±2.5 | 31.6b±1.4[c] | 83.633±1.9 | 7.5±0.01[b] |
| PI372596 | 14.2±0.7[c] | 40.547±0.05[bc] | 0.468±0.002[b] | 1.21±0.03[b] | 0.168±0.01[a] | 0.173±0.004[b] | 0.208±0.01 | 2.4±0.8[b] | 69.767±4.4 | 37.133±2.6[ab] | 70.167±5.2 | 7.8±0.01[b] |

Data are mean±standard error of the mean. Different lowercase letters within a column indicate significant differences at $P<0.05$ as per Tukey's honest significant difference (HSD)test.

**Table 3. Seed nutritional profile and seed yield per plant (SYP) for eight chickpea genotypes under heat stress conditions.**

| Genotype | Protein (%) | Carbon (%) | Phosphorus (%) | Potassium (%) | Calcium (%) | Magnesium (%) | Sulfate (%) | Copper (ppm) | Iron (ppm) | Manganese (ppm) | Zinc (ppm) | SYP (g) |
|---|---|---|---|---|---|---|---|---|---|---|---|---|
| PI368485 | 26.4±2.6[a] | 40.583±0.1[bc] | 0.649±0.01[a] | 1.55±0.03[a] | 0.092±0.003[d] | 0.203±0.003[a] | 0.299±0.001[a] | 2.233±0.5 | 68.43±4.4[ab] | 37.47±2.45[b] | 91.43±0.93[a] | 2.5±0.01[f] |
| PI518255 | 17.1±0.86[c] | 40.19±0.1[bc] | 0.545±0.01[b] | 1.41±0.06[abc] | 0.141±0.003[ab] | 0.174±0.00[cd] | 0.286±0.004[ab] | 2.5±0.5 | 52.66±3.1[bc] | 41.06±1.4[b] | 69.4±2[c] | 4.9±0.12[a] |
| PI360691 | 22.83±0.2[ab] | 40.44±0.2[bc] | 0.556±0.02[b] | 1.29±0.04[c] | 0.123±0.01[bc] | 0.186±0.004[b] | 0.259±0.01[bc] | 3.2±0.2 | 41.53±1.7[c] | 27.23±3[d] | 69.167±2.4[e] | 2.9±0.12[e] |
| PI598080 | 16.4±0.2[c] | 40.58±0.1[bc] | 0.49±0.01[bd] | 1.36±0.03[bc] | 0.164±0.002[a] | 0.168±0.004[d] | 0.239±0.01[c] | 2.567±0.1 | 58.13±1.6[c] | 52.76±2.5[a] | 67.9±2.4[c] | 3.5±0.06[c] |
| PI360688 | 16.3±1.38[c] | 41.18±0.2[a] | 0.442±0.004[d] | 1.51±0.06[ab] | 0.093±0.01[e] | 0.155±0.01[e] | 0.247±0.004[c] | 1.833±0.3 | 68.43±0.5[ab] | 26.9±3.2[d] | 69.03±1.6[c] | 2.7±0.06[ef] |
| Gocke | 18.9±0.8[c] | 41.123±0.2[a] | 0.548±0.02[b] | 1.51±0.03[ab] | 0.104±0.003[cd] | 0.183±0.004[bc] | 0.28±0.01[ab] | 1.733±0.3 | 77.73±1.6[a] | 30.43±0.8[cd] | 84.3±2.7[b] | 3.39±0.06[cd] |
| PI513144 | 19.83±0.9[bc] | 40.72±0.3[ab] | 0.537±0.02[bc] | 1.55±0.04[a] | 0.085±0.004[d] | 0.17±0.003[d] | 0.296±0.01[a] | 2.767±0.2 | 59.1±1.6[b] | 35.83±2.3[bc] | 81.26±3.5[b] | 3.9±0.06[b] |
| PI372596 | 16.9±0.9[c] | 40.12±0.04[c] | 0.547±0.02[b] | 1.31±0.1[c] | 0.137±0.02[ab] | 0.184±0.002[b] | 0.259±0.02[bc] | 2.133±0.6 | 81.3±13.7[a] | 28.53±1.8[d] | 72.9±1.4[c] | 3.2±0.2[d] |

Data are mean±standard error of the mean. Different lowercase letters within a column indicate significant differences at $P<0.05$ as per Tukey's honest significant difference (HSD)test.

**Table 4. Significance of temperature (T), genotype (G), and T×G interactions on seed nutritional profile and seed yield per plant (SYP).**

| Trait | Variables | | | Mean | |
|---|---|---|---|---|---|
| | Treatment(T) | Genotype(G) | T×G | Control | Heat stress |
| Protein (%) | 10.92 | **53.9**** | **20.2**** | 18.4[a] | 19.3[a] |
| Carbon (%) | 0.77 | **0.54**** | **0.2**** | 40.8[a] | 40.6[b] |
| Phosphorus (%) | **0.051**** | **0.013**** | **0.006**** | 0.47[a] | 0.54[b] |
| Potassium(%) | **0.077*** | **0.054 **** | 0.024 | 1.36[a] | 1.44[a] |
| Calcium(%) | 0 | **0.007**** | **0.001**** | 0.121[a] | 0.117[b] |
| Magnesium(%) | **0.001**** | **0.001**** | **0**** | 0.169[a] | 0.178[a] |
| Sulfur(%) | **0.019**** | **0.001*** | **0.001**** | 0.230[a] | 0.271[b] |
| Manganese (ppm) | 50.6 | **284.96**** | **101.6**** | 32.9[a] | 35.01[a] |
| Zinc (ppm) | 6.93 | 119.49 | **236.73*** | 74.9[a] | 75.6[a] |
| Copper (ppm) | **2.85**** | 1.3 | **1.9*** | 2.85[a] | 2.37[a] |
| Iron (ppm) | **1,475.3*** | 550.8 | **828.6**** | 74.5[a] | 63.4[a] |
| SYP(g) | **113.02**** | **5.79**** | **1.73**** | 4.524[a] | 1.779[b] |

Means were separated using Tukey's honest significant difference (HSD) test at $P \leq 0.05$. Different superscripted letters and numbers in bold indicate significant effects on variables or means.

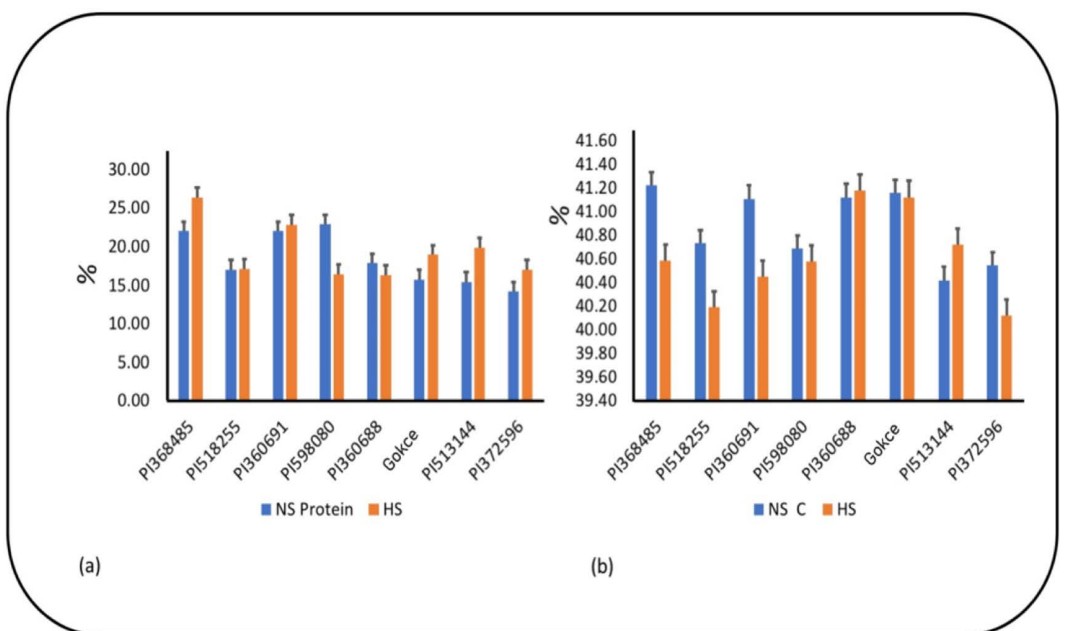

(a)                                    (b)

**Fig 2. (a) Seed protein concentration of chickpea genotypes under non-stress(NS) and heat stress (HS)conditions.** (b) Seed carbon (C) concentration of chickpea genotypes under non-stress(NS) and heat stress (HS)conditions Values are means+SE. ($n=3$). Tukey's test was used to examine treatment differences ($P<0.05$).

effects (Table 4). Deviating from the trend observed in the other genotypes under HS, PI598080 and PI360688 exhibited a decrease in protein concentrations, whereas PI360688 and PI513144 exhibited an increase in seed C (see Table 3).

### Primary and secondary macronutrients

Total seed K had individual G and T effects but no significant G×T interaction (Table 4 and S2 Table in S1 File). Seed P, Mg, and S concentrations exhibited significant T, G, and G×T interaction effects, typically increasing under HS (Figs 3 and 4). However, PI513144 and PI598080 decreased Mg and S concentrations under HS, respectively. Seed Ca had significant treatment and interaction effects, typically decreasing under HS (Fig 3). However, PI 368485 and PI 360691 increased Ca concentration under HS.

### Micronutrients

Seed Cu, Zn, Fe, and Mn concentrations displayed significant genotype (G) × temperature (T) interaction effects (Table 4), with seed Zn and Fe also showing significant temperature effects. Under heat stress (HS), seed Zn and Mn concentrations increased, while seed Fe and Cu concentrations typically decreased (Fig 5). However, counter to the general trend, PI 513144 increased seed Cu concentration, and Gocke, PI 368485, and PI 372596 increased Fe concentrations under HS (Table 3 and Fig 5).

### Seed yield per plant

We observed a significant decrease in seed yield per plant (SYP) (29–66%, Fig 6) due to heat stress, with strong influence from genotype (G), temperature (T), and their interaction (G×T). Notably, PI360691 demonstrated the lowest SYP reduction under heat stress among all tested genotypes, identifying it as a potential candidate for heat tolerance breeding.

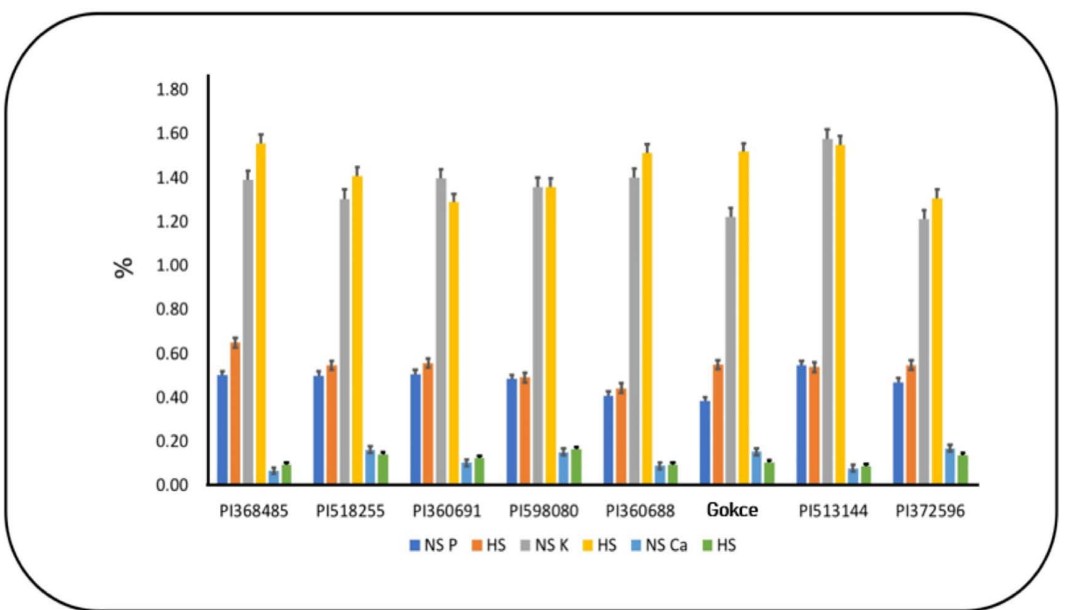

**Fig 3. (a) Seed phosphorus (P),potassium (K), and calcium(Ca) concentrations of chickpea genotypes under non-stress(NS) and heat stress (HS)conditions.** Values are means＋SE ($n=3$). Tukey's test was used to examine treatment differences ($P<0.05$).

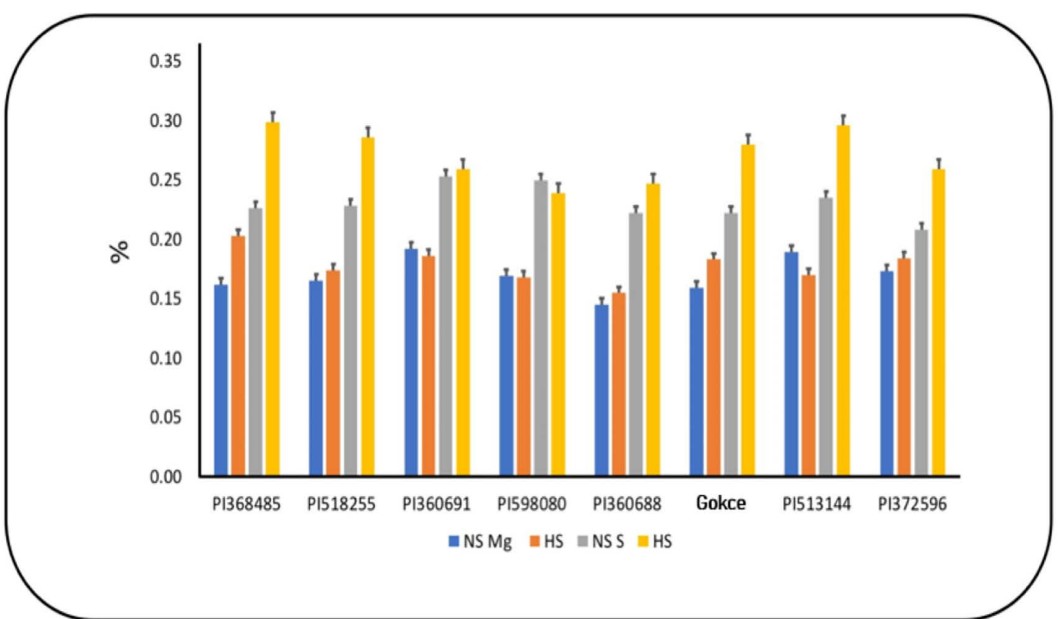

**Fig 4. Seed magnesium (Mg) and sulfur (S) concentrations for chickpea genotypes under non-stress(NS) and (b) heat stress (HS) conditions.** Values are means + SE ($n = 3$). Tukey's test was used to examine treatment differences ($P < 0.05$).

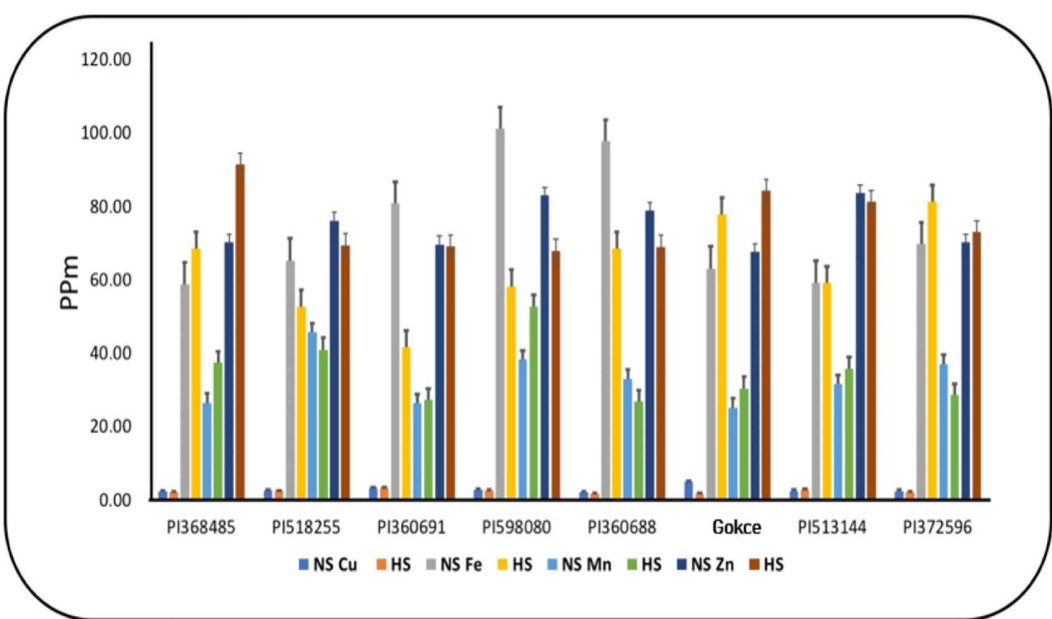

**Fig 5. Seed copper (Cu), iron (Fe), manganese (Mn), and zinc (Zn) concentrations for chickpea genotypes under non-stress(NS) and heat stress (HS) conditions.** Values are means + SE ($n = 3$). Tukey's test was used to examine treatment differences ($P < 0.05$).

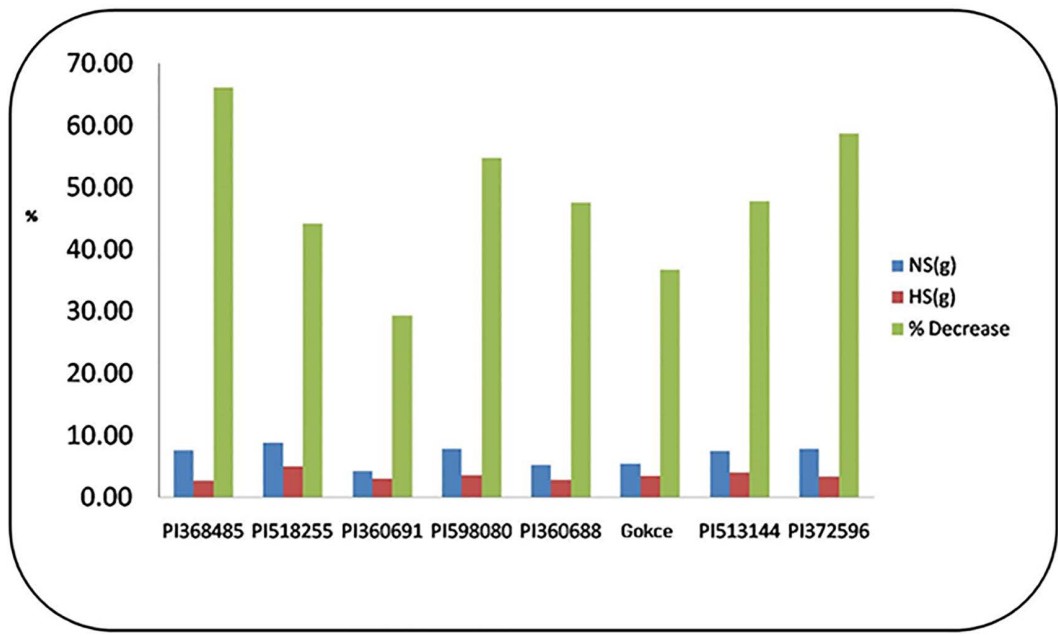

**Fig 6. Percentage reduction in seed yield/plant (g) for eight selected chickpea genotypes under heat stress.** NS = non-stress, HS = heat stress.

### Correlation analysis

Under NS, significant positive association occurred between seed C and protein content (0.44), seed Mg and P content (0.77), and seed S and protein content (0.71) (Fig 7). Seed Ca negatively correlated with K (−0.79).

Under HS, positive association occurred between seed protein and Mg (0.88), seed protein and P (0.82), seed Mg and P (0.89), seed Zn and P (0.71), seed Zn and K (0.72), and seed Zn and S (0.74) (Fig 8), whereas negative association occurred between seed Ca and K (−0.80) and seed Fe and Cu (−0.87).

### Principal component (PC) analysis

Analysis under non-stress (NS) conditions revealed that three PCs collectively accounted for 74.5% of the total variability across the 11 traits (Figs 9 and 11). Individually, PC1 contributed 31.8%, PC2 26.4%, and PC3 16.3%.

Examining the component loadings (Table 5), PC1 was positively driven by P (0.76), K (0.80), Zn (0.63), and S (0.64), with C (−0.64) exhibiting the most pronounced negative contribution. PC2 showed C (0.65) and protein (0.63) as the strongest positive contributors, while Ca (−0.75) was the most negatively influential. For PC3, Fe (0.91) and protein (0.46) were the primary positive contributors, and Mg (−0.52) had the most significant negative impact. PC4 saw Ca (0.56) as its largest positive contributor, contrasted by K (−0.31) as the most significant negative contributor.

Under HS, the initial four principal components (PCs) collectively accounted for 78.8% of the total variability. PC1 contributed 38.7%, PC2 26.7%, and PC3 13.4% (Figs 10 and 11). PC1 was primarily driven by positive contributions from Zn (0.95), protein (0.79), P (0.70), and Mg (0.64), with Ca showing the most significant negative influence (−0.79). For PC2, Cu (0.79) and P (0.65) were the strongest positive contributors, while C exhibited the most notable negative contribution (−0.78) (Table 6). SYP (0.82) and S (0.58) were the main positive drivers for PC3, and Mg had the greatest negative impact (−0.37). Finally, PC4 saw Fe (0.71) as its largest positive contributor, with K showing the greatest negative contribution (0.21).

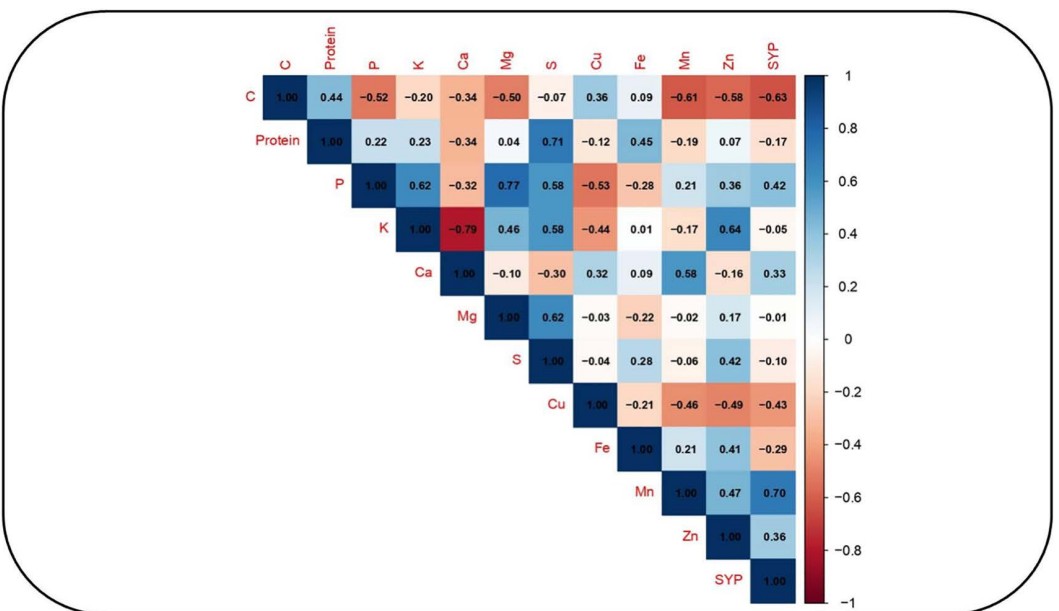

**Fig 7. Correlation analysis forvarious nutritional traits in eight selected chickpea genotypes under non-stress conditions.** Protein = seed protein (%); C = seed carbon (%), P = phosphorous (%), K = potassium (%), Ca = calcium (%), Mg = magnesium (%), S = sulfur (%), Mn = manganese (ppm), Zn = zinc (ppm), Cu = copper (ppm), Fe = iron (ppm), and SYP = seed yield per plant **(g)**.

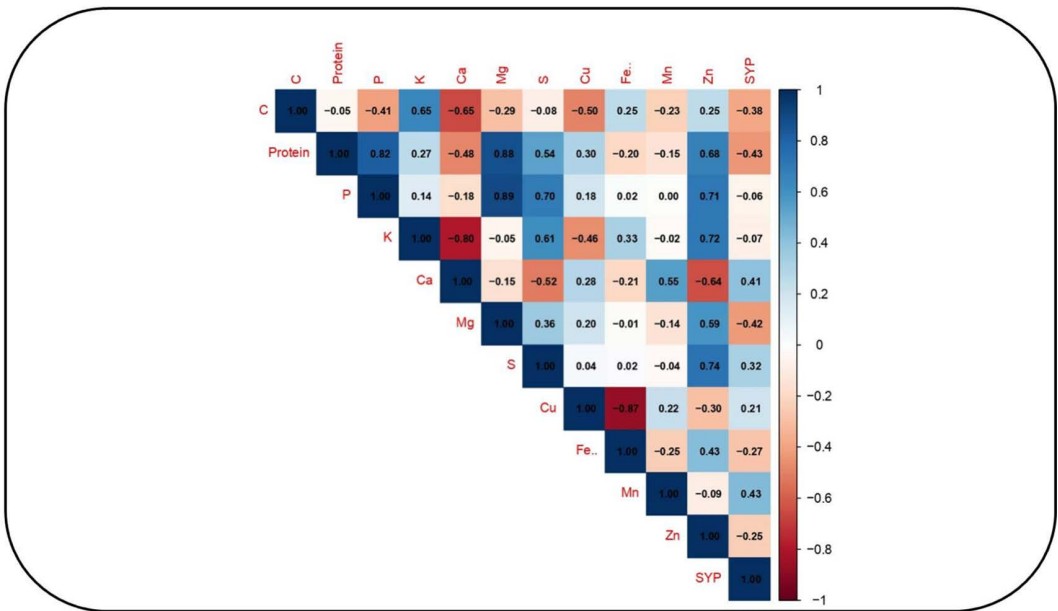

**Fig 8. Correlation analysis forvarious nutritional traits in eight selected chickpea genotypes under heat stress conditions.** Protein = seed protein (%); C = seed carbon (%), P = phosphorous (%), K = potassium (%), Ca = calcium (%), Mg = magnesium (%), S = sulfur (%), Mn = manganese (ppm), Zn = zinc (ppm), Cu = copper (ppm), Fe = iron (ppm), and SYP = seed yield per plant **(g)**.

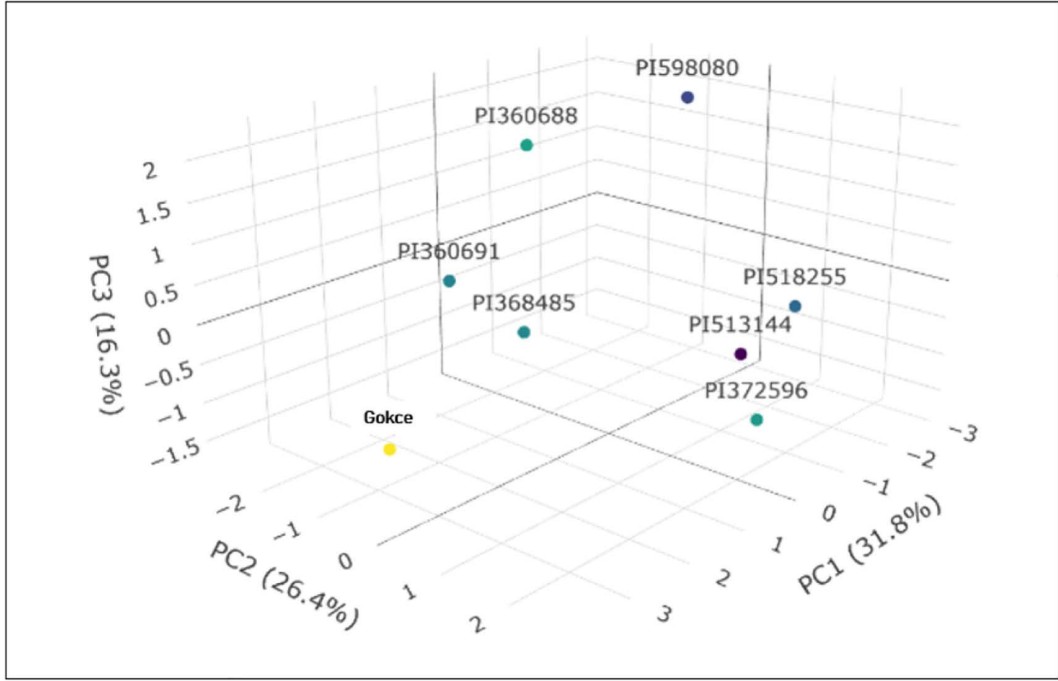

**Fig 9. Principal component analysis (PCA) for various seed nutritional traits in eight selected chickpea genotypes under non-stress conditions.**

## Clustering analysis

Under NS, the genotypes formed two main clusters: (1) Gocke, PI368485, PI513144, PI518255, PI372596, and PI360691 and (2) PI598080 and PI360688 (Fig 12). Under HS, the genotypes also grouped into two clusters: (1) PI360688, Gokce, PI372596, PI513144, and PI368485 and (2) PI360691, PI518255, and PI598080 (Fig 13).

## Discussion

Heat stress significantly affects crop yields and is exacerbated by significant genotype × temperature (G × T) interactions [6,11]. Heat stress also alters the seed macronutrient and micronutrient components in various grain legumes, including chickpea [18,22,28,29].

Our study revealed that heat stress in chickpea decreased seed C concentration but increased seed protein concentration. Similar reductions in seed carbohydrates due to HS have been reported in maize (*Zea mays* L.), with declines of 3.0% in 2014 and 3.3% in 2015 [30], and in rice (*Oryza sativa* L.), where HS disrupted sucrose synthase and starch branching enzyme activities, hindering starch accumulation [31]. In wheat (*Triticum aestivum* L.), HS inhibits key enzymes involved in starch metabolism, such as soluble starch synthase, sucrose synthase, glucokinase, and ADP glucose pyrophosphorylase, reducing carbohydrate levels [32]. Similar trends have been reported in grain legumes such as chickpea [18,21,23,29]. The decline in carbohydrate concentration is likely due to reduced enzymatic activity in starch and sucrose synthesis, including sucrose phosphate synthase, soluble starch synthase, and sucrose synthase [18,30], coupled with restricted sucrose transport from leaves to developing pods and seeds [31].

In contrast to carbohydrate depletion, seed protein concentration often increases under HS, as reported in wheat [33,34], maize (24.5% in 2014 and 25.3% in 2015) [30], hard fescue (*Festuca trachyphylla*) [35], and soybean (*Glycine max* L.) [36]. This increase is attributed to enhanced enzyme activity, including "*glutamate synthase and glutamate*

**Table 5. Correlation between variables and PCs under non-stress conditions.**

| Variables | PC1 | PC2 | PC3 | PC4 | PC5 | PC6 | PC7 |
|---|---|---|---|---|---|---|---|
| Carbon (%) | −0.64 | 0.65 | 0.17 | −0.13 | 0.281 | 0.042 | 0.203 |
| Protein (%) | 0.23 | 0.63 | 0.46 | 0.27 | 0.501 | 0.072 | −0.09 |
| Phosphorus (%) | 0.9 | 0.05 | −0.35 | 0.07 | 0.28 | −0.139 | 0.008 |
| Potassium (%) | 0.8 | 0.48 | −0.1 | −0.31 | −0.246 | 0.122 | 0.069 |
| Calcium (%) | −0.32 | −0.75 | 0.13 | 0.56 | 0.048 | −0.028 | −0.017 |
| Magnesium (%) | 0.63 | 0.15 | −0.52 | 0.49 | −0.148 | −0.236 | 0.014 |
| Sulfur (%) | 0.64 | 0.53 | 0.12 | 0.52 | 0.063 | 0.158 | 0.064 |
| Copper(ppm) | −0.64 | 0.17 | −0.29 | 0.52 | −0.232 | 0.38 | 0.032 |
| Iron (ppm) | 0.07 | 0.16 | 0.91 | 0.19 | −0.25 | −0.213 | −0.094 |
| Manganese(ppm) | 0.37 | −0.79 | 0.37 | 0.102 | 0.109 | −0.05 | 0.291 |
| Zinc (ppm) | 0.76 | −0.14 | 0.41 | −0.13 | −0.383 | 0.291 | 0.003 |
| Seed yield per plant(g) | 0.43 | −0.72 | −0.03 | −0.11 | 0.409 | 0.312 | −0.113 |

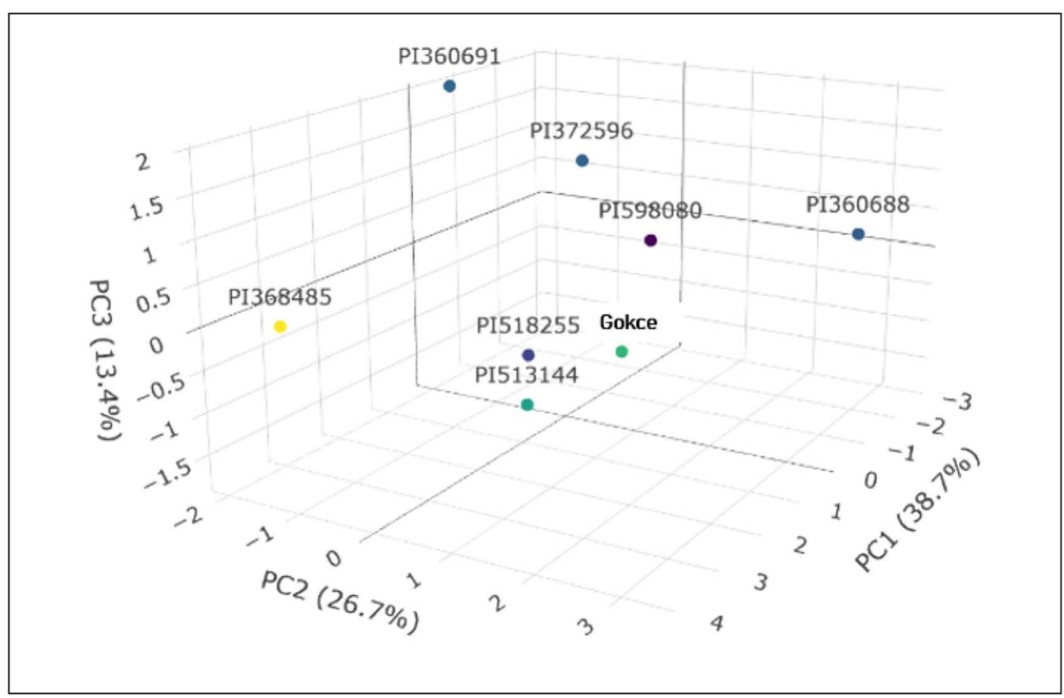

**Fig 10. Principal component analysis (PCA) for various seed nutritional traits in eight selected chickpea genotypes under heat stress conditions.**

*pyruvate transaminase*", along with decreased glutamine synthetase activity during grain filling [30]. Additionally, increased cellular production of heat shock proteins, heat shock factors, and stress signaling proteins, which help plants adapt to HS, contribute to this increase in protein [30,37]. However, HS has been reported to decrease seed protein concentrations in crops such as *Vigna radiata* [29,37], *Cicer arietinum* [18,24,38], soybean [37], *Lens culinaris* [22], and *Lathyrus sativus* L [28].

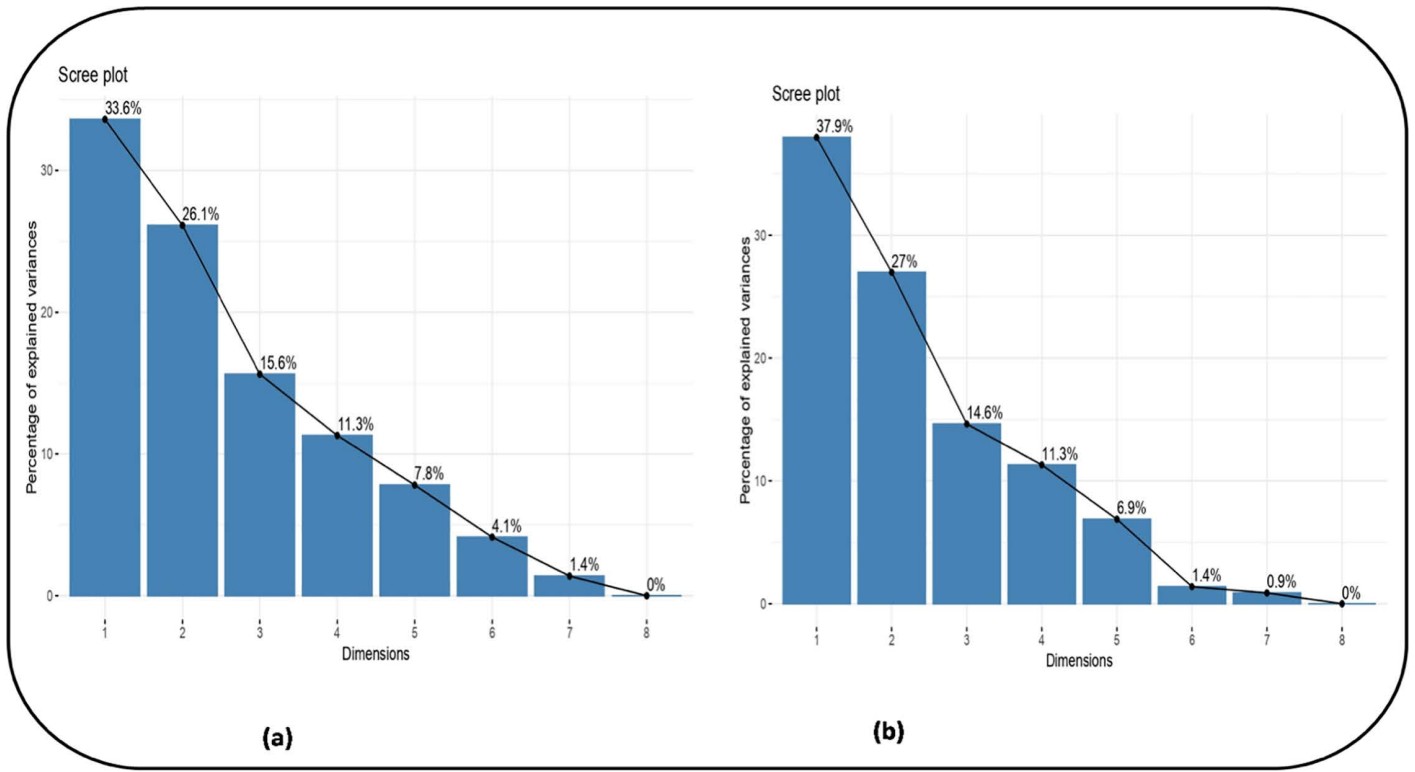

**Fig 11. Scree plot constructed using eightprincipal components for eight selected chickpea genotypes under (a) non-stress and (b) heat stress conditions.**

**Table 6. Correlation between variables and PCs under heat stress conditions.**

| Variables | PC1 | PC2 | PC3 | PC4 | PC5 | PC6 | PC7 |
|---|---|---|---|---|---|---|---|
| Carbon (%) | 0.3 | −0.78 | −0.04 | −0.4 | 0.259 | 0.226 | 0.122 |
| Protein (%) | 0.79 | 0.52 | −0.23 | −0.19 | 0.144 | 0.009 | −0.074 |
| Phosphorus (%) | 0.701 | 0.65 | 0.05 | 0.27 | −0.034 | 0.068 | −0.02 |
| Potassium (%) | 0.7 | −0.47 | 0.47 | −0.21 | 0.149 | −0.081 | −0.086 |
| Calcium (%) | −0.79 | 0.39 | 0.02 | 0.43 | 0.168 | 0.092 | 0.03 |
| Magnesium (%) | 0.64 | 0.6 | −0.37 | 0.22 | 0.104 | 0.145 | 0.014 |
| Sulfur (%) | 0.71 | 0.27 | 0.58 | −0.05 | −0.281 | −0.028 | −0.014 |
| Copper (ppm) | −0.25 | 0.79 | −0.05 | −0.5 | −0.009 | −0.159 | 0.193 |
| Iron (ppm) | 0.32 | −0.6 | −0.04 | 0.71 | −0.054 | −0.127 | 0.109 |
| Manganese (ppm) | −0.32 | 0.29 | 0.54 | 0.12 | 0.703 | −0.059 | −0.024 |
| Zinc (ppm) | 0.95 | 0.005 | 0.19 | 0.17 | 0.128 | −0.019 | 0.154 |
| Seed yield per plant (g) | −0.4 | 0.23 | 0.82 | 0.04 | −0.288 | 0.169 | 0.049 |

Our study also found significant T, G, and G × T interaction effects for seed P, Mg, Ca, and S concentrations, which increased under HS. Phosphorus is critical for mitigating HS by influencing seed metabolism, reducing seed oil and starch concentrations [39], and enhancing seed quality traits; it is also essential in mediating mitogen-activated protein kinase cascades during HS [40]. Magnesium, vital for chlorophyll synthesis and carbon metabolism, typically increases under

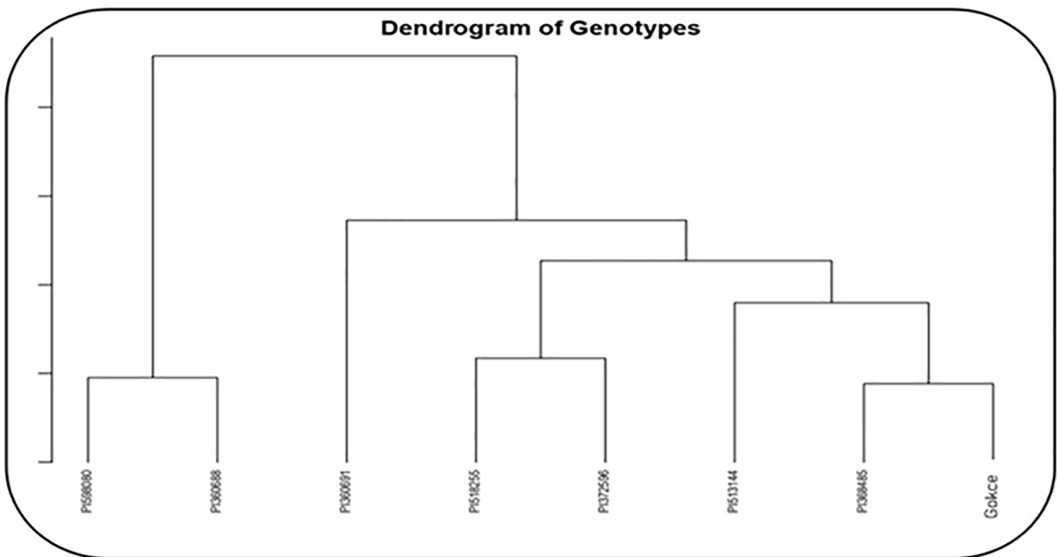

**Fig 12. Cluster analysis of eight selected chickpea genotypes under non-stress conditions.**

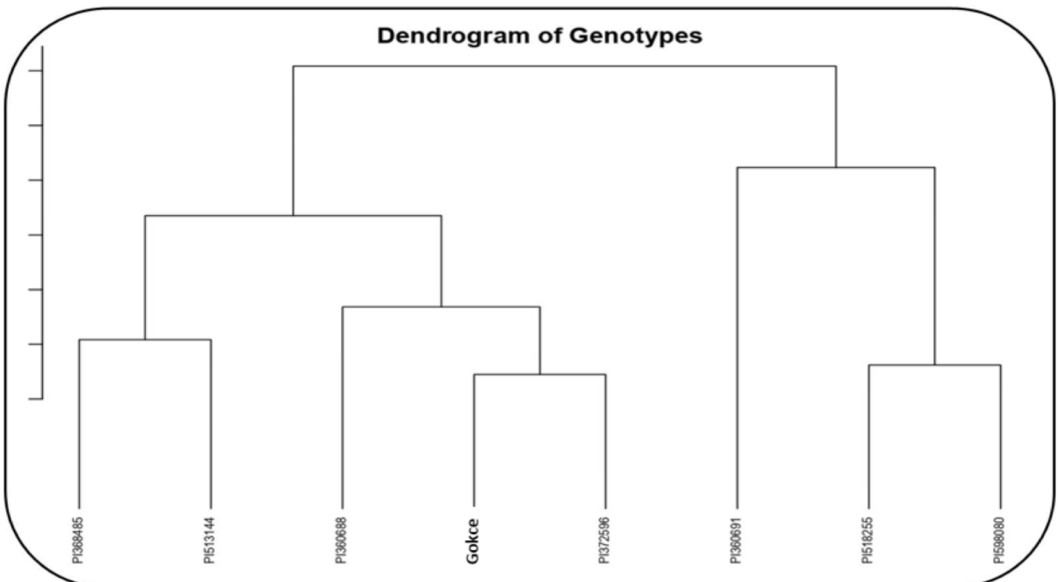

**Fig 13. Cluster analysis of eight selected chickpea genotypes under heat stress conditions.**

heat stress, aiding plant tolerance [41]. However, decreased seed Mg, Ca, and K concentrations have been noted in mung bean [29] and lentil [21] under similar conditions. In this study, seed Ca concentration decreased under HS, likely due to its role as an intracellular signaling molecule in HS responses [42]. In contrast, seed S concentration increased, which may be linked to its role in synthesizing methionine—an amino acid crucial for heat shock protein function [43]—and its involvement in maintaining cellular redox balance and scavenging reactive oxygen species (ROS) [44,45]. Notably, exogenous S application alleviated heat stress in tomato plants by increasing proline, N, P, and K contents [46].

Micronutrient concentrations also shifted under HS, with a decline in seed Fe levels, consistent with previous reports in chickpea [17,18,24], lentil [21–23,47], and mung bean [29]. Conversely, seed Zn concentration increased, which may enhance C assimilation and alleviate oxidative stress through antioxidant production [48]. Similarly, Mn levels increased in chickpea under HS [17], potentially contributing to improved ROS scavenging and upregulation of chaperone proteins, enhancing HS tolerance [49,50]. However, reductions in seed Zn concentration under HS have also been reported in chickpea [17,24], lentil [21,22], and mung bean [29], highlighting potential genotype-dependent responses.

Association analysis revealed significant positive associations between seed Mg and P concentrations under NS conditions, consistent with findings in other legumes, such as soybean, pea, lentil, and beans [51], and sesame (*Sesamum indicum* L.) [52]. A strong positive associationalso occurred between seed protein and S concentrations, as reported in chickpea [53,54]. Conversely, seed Ca and K concentrations negatively correlated, consistent with the results in sesame [52].

Significant positive correlations were observed under HS between seed P and Zn concentrations, and between seed protein and Mg concentrations, consistent with previous studies on chickpea [55], winter oilseed rape [56], and soybean [57]. A positive correlation between seed Zn and S concentrations was also noted, aligning with reports in wheat [58]. Conversely, significant negative correlations occurred under HS for seed Ca and K concentrations, and for seed Fe and Cu concentrations. While the negative correlation between seed Ca and K is in agreement with findings in sesame [52], the negative correlation between seed Fe and Cu diverges from a reported positive correlation in the same sesame study.

Clustering analysis grouped the genotypes into two distinct clusters under NS and HS conditions. Distant genotypes within each cluster could be valuable for breeding programs aimed at developing nutrient-rich, heat-tolerant chickpea varieties. Similar clustering-based selection strategies have been successfully applied in chickpea [59], lentil [60], and common bean [61]. It is thought that the chickpea genotypes used in this study have different heat stress thresholds, as in Phaseolus species having different heat stress tolerances or sensitivities [62], which affect the variation of some macro and micronutrient contents.

## Conclusions

Heat stress at 35/20°C significantly reduced chickpea yield and adversely affected seed quality, including decreased C, Ca, Cu and Fe concentrations. However, seed protein and P, K, S, Mg, Mn, and Zn concentrations were either less affected or increased under HS. The positive correlations observed between seed protein and both Mg and P suggest that enhancing protein levels could also improve other nutritional traits under HS. Therefore, maintaining an optimal balance of seed C, protein, and other essential nutrients could help meet chickpea's calorific and nutritional demandsas temperatures continue to rise. Future research focusing on the molecular mechanism underlying the reductions in C, Ca, Mg, and Fe concentrations under HS could provide valuable insights into improving these nutrient contents in chickpea under HS conditions. Chickpeas with higher nutritional values under HS conditions could be potentially used in breeding programs to improve chickpea seed nutritional quality.

## Supporting information

**S1 File. S1 Fig. Growth chamber temperatures recorded during chickpea growth. S1 Table Chickpea (*Cicer arietinum* L.) accessions used in the study. S2 Table Combined/ interaction effect of eight chickpea genotypes under non stress and heat stress on seed nutrition components.**
(DOCX)

## Acknowledgments

UCJ and PV acknowledge support from KSU. We also thank Eric J. Bishop von Wettberg for providing seed of Gokce chickpea genotype.

## Author contributions

**Conceptualization:** Uday Jha, Harsh Nayyar, P.V. Vara Prasad.

**Formal analysis:** Uday Jha, Sadiah Shafi.

**Investigation:** P.V. Vara Prasad.

**Methodology:** Uday Jha, Ignacio A. Ciampitti.

**Resources:** Marilyn Warburton, P.V. Vara Prasad.

**Software:** Uday Jha, Marilyn Warburton, Harsh Nayyar, Sadiah Shafi, Ashis Ranjan Udgata.

**Supervision:** P.V. Vara Prasad.

**Visualization:** P.V. Vara Prasad.

**Writing – original draft:** Uday Jha.

**Writing – review & editing:** Marilyn Warburton, Harsh Nayyar, Sadiah Shafi, Ignacio A. Ciampitti, Kadambot H.M. Siddique, P.V. Vara Prasad.

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
