## [Decision Letter · Decision Letter 0]

3 Jul 2025

Dear Dr. Jha,

Thank you for submitting your manuscript to PLOS ONE. After careful consideration, we feel that it has merit but does not fully meet PLOS ONE’s publication criteria as it currently stands. Therefore, we invite you to submit a revised version of the manuscript that addresses the points raised during the review process.

We look forward to receiving your revised manuscript.

Kind regards,

Prafull Salvi, PhD

Academic Editor

PLOS ONE

Journal Requirements:

3. Please amend the manuscript submission data (via Edit Submission) to include author Uday Chand Jha.

4. Please amend your authorship list in your manuscript file to include author Uday Jha.

6. Please remove all personal information, ensure that the data shared are in accordance with participant consent, and re-upload a fully anonymized data set.

Additional Editor Comments :

Reviewer 1:

This is very good study on heat stress tolerance in chickpea. The results are interesting and presented/discussed in details.

It may be accepted for publication after a few minor edits/corrections which are mentioned below:

1. Abstract-8th row: what is “seed (S%)’ ? Not clear. Please make correction, if required.

2. Page 9-5th row: In this sentence "As a cool-season legume, chickpea is well adapted to mild-temperature environments but highly susceptible to heat stress during vegetative and reproductive stages3" what does “stages3” indicate? Is it a reference [3] or something else ? Not clear'

3. Please check name of kabuli chickpea variety “Gokce”. Different and incorrect spellings of name of this variety are written in text, Tables and Figures. The correct name is "Gokce". Please verify check the correct name and replace in whole MS.

4. Page 10: In this sentence “The experiment was conducted under NS conditions at Kansas State University, Manhattan, in 2023–2024” but there is no mention about HS experiment, please check and make necessary corrections.

5. Page 16-first para: In this sentence “However, HS has been reported to decrease seed protein concentrations in crops such as mung bean [30,38], chickpea [18,24,39], soybean [38], Lens culinaris [22], and Lathyrus

sativus L[29]” keep uniformity for writing name of crop -write either common name or scientific name.

6. Table 1: CV for Carbon (C) is very low, i.e. 0.76 for NS and 0.95 for HS, please check.

7. Table 2: Concentration of Manganese (ppm) is very low, i.e. 6.5, please check again. It should be in between 30 and 40.

8. Table 5: Value of PC5 for Calcium (%) is ‘0’, please check.

Reviewer 2:

Dear author(s),

I have reviewed yor manuscript (PONE-D-25-29542) "Influence of Elevated Temperature on the Nutritional Profile of Chickpea (Cicer arietinum L.) Seeds". My comments and suggestions are provided below:

Title

1. Seeds in the title can be removed since seeds are used for nutrition in chickpea. Alternative title: Influence of Heat Stress on the Nutritional Profile of Chickpea (Cicer arietinum L.)

Abstract

2. Change “those of chickpea” to “nutritional profile of chickpea”

3. A blank between effect and of

4. Change “cultivating eight chickpea genotypes” to “eight cultivated chickpea genotypes”

5. The following sentence must not be italicized seed carbon (C, %), protein (%), phosphorus (P, %), potassium (K, %), magnesium (Mg, %), sulfur (S, %), and manganese (Mn, ppm)

6. Change “seed (S%), seed iron (Fe, ppm),” to “sulfur (S%), iron (Fe, ppm),”. Readers are already aware that the edible portion of cultivated chickpeas is the seed.

7. Change “seed copper (Cu, ppm),” to “copper (Cu, ppm),”.

8. Change “seed C” to “C”

9. Change “seed M” to “Mg”

10. Change “seed protein” to “protein”

11. Please check others.

Introduction

12. Before reference [2] (BR2), FAOSTAT data should be updated, and the 2023 data (available). Write in 2023 in the sentence for readers.

13. BR3, The reference 3 should be written properly. Please avoid self-citation.

14. BR3-6, India is the largest chickpea-producing country; however, the given samples on heat stress (˃32°C) may be representative of a broader context (in the world).

15. BR12, Read the articles below for more and different references: https://doi.org/10.3390/agronomy12030557;
https://doi.org/10.1038/s41598-022-05559-3.

16. BR13-14, a blank between that and temperatures

17. BR15, a blank between yield and by

18. BR17-19, a blank before [17-19]

19. BR20, a blank at the end of the sentence.

20. BR19, a blank before [19]

21. BR24, a blank before [24]

22. BR18, a blank between with and heat-sensitive

23. A blank between nutrition and traits

24. A blank between into and developing

25. Please use more references on chickpea.

M&M

Plant materials

26. Check Gockce? Why did you use one cultivar?

Experimental setup

27. What was the irrigation quantity?

28. A blank before non-stress

29. Some traits of the soil used in the experiment should be detailed.

30. Sowing, planting and harvesting dates should be given.

31. Change “Data analysis” to Data analyses”. There was more than one analysis.

32. Duncan or Tukey tests could be used instead of LSD.

Results

33. Remove soil from “Under normal soil conditions,” because NS was explained as non-stress (NS) conditions.

34. Remove the seed as well. You explained seed analyses before for readers.

35. Write conditions after NS

36. Change “under high salinity (HS) conditions” to “under heat stress (HS) conditions”

37. Check and correct seed yield per plant (SYP) under heat-stress (HS) conditions! 34.9 is higher than NS.

Protein and carbon concentration

38. Change “decreased protein concentrations” to “a decrease in protein concentrations”

39. Change “increased seed C” to “an increase in seed C”

40. Remove “seed” because you mentioned that analyses were made in the seed in M&M.

Micronutrients

41. Change “Gockce” to “Gokce”.

Seed yield per plant

42. Remove “identifying it as a potential candidate for heat tolerance breeding” from here to “Discussion”.

43. Abbreviations (G, T, HS, NS etc) were repeated. Please insist on full name or abbreviations through text.

Discussion

44. BR37, a blank before [37].

45. A blank between to and HS

46. BR39, a blank before has.

47. The scientific name of mung bean can be written.

48. The common names of Lentil and Lathyrus should be given.

49. The following sentence could be written as reference [63] “It is thought that the chickpea genotypes used in this study have different heat stress thresholds, as in Phaseolus species having different heat stress tolerances or sensitivities [63], which affect the variation of some macro and micronutrient contents.” 63. Tene et al., Star of biochemical traits of heat-tolerant and heat-sensitive Phaseolus genotypes in coping with heat stress. Plant Growth Regulation. 2025; doi: https://doi.org/10.1007/s10725-025-01340-4

Conclusion

50. The following sentence or similar sentence could be added in Conclusion, too. Chickpeas having the higher nutritional values under HS conditions were suggested to use in breeding programs.

References

51. All references should be checked and corrected according to “Guide for Authors”.

52. The scientific names of plant species should be italicized.

I hope you will find as useful my suggestion to improve your mn. Please use line numbers. Review was taken more time than I thought due to without line no.

Reviewers' comments:

Reviewer's Responses to Questions

**Comments to the Author**

1. Is the manuscript technically sound, and do the data support the conclusions?

Reviewer #1: Yes

Reviewer #2: Yes

2. Has the statistical analysis been performed appropriately and rigorously?

Reviewer #1: Yes

Reviewer #2: Yes

3. Have the authors made all data underlying the findings in their manuscript fully available?

Reviewer #1: Yes

Reviewer #2: Yes

4. Is the manuscript presented in an intelligible fashion and written in standard English?

Reviewer #1: Yes

Reviewer #2: Yes

Reviewer #1: This is very good study on heat stress tolerance in chickpea. The results are interesting and presented/discussed in details.

It may be accepted for publication after a few minor edits/corrections which are mentioned below:

1. Abstract-8th row: what is “seed (S%)’ ? Not clear. Please make correction, if required.

2. Page 9-5th row: In this sentence "As a cool-season legume, chickpea is well adapted to mild-temperature environments but highly susceptible to heat stress during vegetative and reproductive stages3" what does “stages3” indicate? Is it a reference [3] or something else ? Not clear'

3. Please check name of kabuli chickpea variety “Gokce”. Different and incorrect spellings of name of this variety are written in text, Tables and Figures. The correct name is "Gokce". Please verify check the correct name and replace in whole MS.

4. Page 10: In this sentence “The experiment was conducted under NS conditions at Kansas State University, Manhattan, in 2023–2024” but there is no mention about HS experiment, please check and make necessary corrections.

5. Page 16-first para: In this sentence “However, HS has been reported to decrease seed protein concentrations in crops such as mung bean [30,38], chickpea [18,24,39], soybean [38], Lens culinaris [22], and Lathyrus

sativus L[29]” keep uniformity for writing name of crop -write either common name or scientific name.

6. Table 1: CV for Carbon (C) is very low, i.e. 0.76 for NS and 0.95 for HS, please check.

7. Table 2: Concentration of Manganese (ppm) is very low, i.e. 6.5, please check again. It should be in between 30 and 40.

8. Table 5: Value of PC5 for Calcium (%) is ‘0’, please check.

Reviewer #2: Dear author(s),

I have reviewed yor manuscript (PONE-D-25-29542) "Influence of Elevated Temperature on the Nutritional Profile of Chickpea (Cicer arietinum L.) Seeds". My comments and suggestions are provided below:

Title

1. Seeds in the title can be removed since seeds are used for nutrition in chickpea. Alternative title: Influence of Heat Stress on the Nutritional Profile of Chickpea (Cicer arietinum L.)

Abstract

2. Change “those of chickpea” to “nutritional profile of chickpea”

3. A blank between effect and of

4. Change “cultivating eight chickpea genotypes” to “eight cultivated chickpea genotypes”

5. The following sentence must not be italicized seed carbon (C, %), protein (%), phosphorus (P, %), potassium (K, %), magnesium (Mg, %), sulfur (S, %), and manganese (Mn, ppm)

6. Change “seed (S%), seed iron (Fe, ppm),” to “sulfur (S%), iron (Fe, ppm),”. Readers are already aware that the edible portion of cultivated chickpeas is the seed.

7. Change “seed copper (Cu, ppm),” to “copper (Cu, ppm),”.

8. Change “seed C” to “C”

9. Change “seed M” to “Mg”

10. Change “seed protein” to “protein”

11. Please check others.

Introduction

12. Before reference [2] (BR2), FAOSTAT data should be updated, and the 2023 data (available). Write in 2023 in the sentence for readers.

13. BR3, The reference 3 should be written properly. Please avoid self-citation.

14. BR3-6, India is the largest chickpea-producing country; however, the given samples on heat stress (˃32°C) may be representative of a broader context (in the world).

15. BR12, Read the articles below for more and different references: https://doi.org/10.3390/agronomy12030557;
https://doi.org/10.1038/s41598-022-05559-3.

16. BR13-14, a blank between that and temperatures

17. BR15, a blank between yield and by

18. BR17-19, a blank before [17-19]

19. BR20, a blank at the end of the sentence.

20. BR19, a blank before [19]

21. BR24, a blank before [24]

22. BR18, a blank between with and heat-sensitive

23. A blank between nutrition and traits

24. A blank between into and developing

25. Please use more references on chickpea.

M&M

Plant materials

26. Check Gockce? Why did you use one cultivar?

Experimental setup

27. What was the irrigation quantity?

28. A blank before non-stress

29. Some traits of the soil used in the experiment should be detailed.

30. Sowing, planting and harvesting dates should be given.

31. Change “Data analysis” to Data analyses”. There was more than one analysis.

32. Duncan or Tukey tests could be used instead of LSD.

Results

33. Remove soil from “Under normal soil conditions,” because NS was explained as non-stress (NS) conditions.

34. Remove the seed as well. You explained seed analyses before for readers.

35. Write conditions after NS

36. Change “under high salinity (HS) conditions” to “under heat stress (HS) conditions”

37. Check and correct seed yield per plant (SYP) under heat-stress (HS) conditions! 34.9 is higher than NS.

Protein and carbon concentration

38. Change “decreased protein concentrations” to “a decrease in protein concentrations”

39. Change “increased seed C” to “an increase in seed C”

40. Remove “seed” because you mentioned that analyses were made in the seed in M&M.

Micronutrients

41. Change “Gockce” to “Gokce”.

Seed yield per plant

42. Remove “identifying it as a potential candidate for heat tolerance breeding” from here to “Discussion”.

43. Abbreviations (G, T, HS, NS etc) were repeated. Please insist on full name or abbreviations through text.

Discussion

44. BR37, a blank before [37].

45. A blank between to and HS

46. BR39, a blank before has.

47. The scientific name of mung bean can be written.

48. The common names of Lentil and Lathyrus should be given.

49. The following sentence could be written as reference [63] “It is thought that the chickpea genotypes used in this study have different heat stress thresholds, as in Phaseolus species having different heat stress tolerances or sensitivities [63], which affect the variation of some macro and micronutrient contents.” 63. Tene et al., Star of biochemical traits of heat-tolerant and heat-sensitive Phaseolus genotypes in coping with heat stress. Plant Growth Regulation. 2025; doi: https://doi.org/10.1007/s10725-025-01340-4

Conclusion

50. The following sentence or similar sentence could be added in Conclusion, too. Chickpeas having the higher nutritional values under HS conditions were suggested to use in breeding programs.

References

51. All references should be checked and corrected according to “Guide for Authors”.

52. The scientific names of plant species should be italicized.

I hope you will find as useful my suggestion to improve your mn. Please use line numbers. Review was taken more time than I thought due to without line no.

**Do you want your identity to be public for this peer review?** For information about this choice, including consent withdrawal, please see our Privacy Policy

Reviewer #1: **Yes: ** Sarvjeet Singh

Reviewer #2: **Yes: ** Cengiz TOKER

---

## [Author Response · Author response to Decision Letter 1]

18 Jul 2025

Dear Editor

We thanked the both reviewers for their valuable suggestions for improving the manuscript. We have addressed the queries of both reviewers. Yellow highlights have been used for addressing correction suggested by reviewer1 and green highlights have been used for addressing corrections suggested by reviewer2. We hope now the manuscript can be accepted. We are also open to revise any more queries left.

Thanking You Sincerely

Uday

Reviewer1

1. Abstract-8th row: what is “seed (S%)’ ? Not clear. Please make correction, if required.

Response: We have corrected it highlighted in yellow in the manuscript.

2. Page 9-5th row: In this sentence "As a cool-season legume, chickpea is well adapted to mild-temperature environments but highly susceptible to heat stress during vegetative and reproductive stages3" what does “stages3” indicate? Is it a reference [3] or something else ? Not clear'

Response: Yes it is reference we have corrected it

3. Please check name of kabuli chickpea variety “Gokce”. Different and incorrect spellings of name of this variety are written in text, Tables and Figures. The correct name is "Gokce". Please verify check the correct name and replace in whole MS.

Response: We have corrected the spelling of kabuli genotype highlighted in yellow in Tables and in manuscript.

4. Page 10: In this sentence “The experiment was conducted under NS conditions at Kansas State University, Manhattan, in 2023–2024” but there is no mention about HS experiment, please check and make necessary corrections.

Response: We have corrected it and mentioned it in Material and method section highlighted in yellow.

5. Page 16-first para: In this sentence “However, HS has been reported to decrease seed protein concentrations in crops such as mung bean [30,38], chickpea [18,24,39], soybean [38], Lens culinaris [22], and Lathyrus

sativus L[29]” keep uniformity for writing name of crop -write either common name or scientific name.

Response: We have written the scientific name for uniformity please see the yellow highlight in the manuscript

6. Table 1: CV for Carbon (C) is very low, i.e. 0.76 for NS and 0.95 for HS, please check.

Thanks for this observation we have checked it but due to very low differences of this value in among the genotypes under both condition.

7. Table 2: Concentration of Manganese (ppm) is very low, i.e. 6.5, please check again. It should be in between 30 and 40.

We have corrected the typo mistake it was 26.5 highlighted in yellow in the table2.

8. Table 5: Value of PC5 for Calcium (%) is ‘0’, please check.

We have corrected the typo it was 0.048 we have corrected it highlighted in yellow in Table5.

Reviewer 2:

Dear author(s),

I have reviewed yor manuscript (PONE-D-25-29542) "Influence of Elevated Temperature on the Nutritional Profile of Chickpea (Cicer arietinum L.) Seeds". My comments and suggestions are provided below:

Title

1. Seeds in the title can be removed since seeds are used for nutrition in chickpea. Alternative title: Influence of Heat Stress on the Nutritional Profile of Chickpea (Cicer arietinum L.)

We have changed the title as suggested by reviewer

Abstract

2. Change “those of chickpea” to “nutritional profile of chickpea”

We have changed it accordingly highlighted in green

3. A blank between effect and of

We have given the space accordingly highlighted in green

4. Change “cultivating eight chickpea genotypes” to “eight cultivated chickpea genotypes”

We have changed it accordingly highlighted in green.

5. The following sentence must not be italicized seed carbon (C, %), protein (%), phosphorus (P, %), potassium (K, %), magnesium (Mg, %), sulfur (S, %), and manganese (Mn, ppm)

We have written the sentence without italicized highlighted in green.

6. Change “seed (S%), seed iron (Fe, ppm),” to “sulfur (S%), iron (Fe, ppm),”. Readers are already aware that the edible portion of cultivated chickpeas is the seed.

We have changed it accordingly please see highlighted in green

7. Change “seed copper (Cu, ppm),” to “copper (Cu, ppm),”.

We have changed it accordingly please see highlighted in green

8. Change “seed C” to “C”

We have changed it accordingly please see highlighted in green

9. Change “seed M” to “Mg”

We have changed it accordingly please see highlighted in green

10. Change “seed protein” to “protein”

We have changed it accordingly please see highlighted in green

11. Please check others.

Introduction

12. Before reference [2] (BR2), FAOSTAT data should be updated, and the 2023 data (available). Write in 2023 in the sentence for readers.

We have updated the reference FAOSTAT 2023

13. BR3, The reference 3 should be written properly. Please avoid self-citation.

We have removed self citation and replaced with other relevant citation.

14. BR3-6, India is the largest chickpea-producing country; however, the given samples on heat stress (˃32°C) may be representative of a broader context (in the world).

We have revised it

15. BR12, Read the articles below for more and different references: https://doi.org/10.3390/agronomy12030557;
https://doi.org/10.1038/s41598-022-05559-3.

We have mentioned the above reference see highlighted in green in reference section

16. BR13-14, a blank between that and temperatures

We have given the space accordingly see highlighted in green

17. BR15, a blank between yield and by

We have given the space accordingly see highlighted in green

18. BR17-19, a blank before [17-19]

We have given the space accordingly see highlighted in green

19. BR20, a blank at the end of the sentence.

We have given the space accordingly please see highlighted in green

20. BR19, a blank before [19]

We have given the space accordingly see highlighted in green

21. BR24, a blank before [24]

We have given the space accordingly see highlighted in green

22. BR18, a blank between with and heat-sensitive

We have given the space accordingly see highlighted in green

23. A blank between nutrition and traits

We have given the space accordingly see highlighted in green

24. A blank between into and developing

We have given the space accordingly see highlighted in green

25. Please use more references on chickpea.

We have used references of chickpea appropriately as in this case reference in chickpea is very limeited

M&M

Plant materials

26. Check Gockce? Why did you use one cultivar?

We have corrected the spelling accordingly see highlighted in green/ yellow

Experimental setup

27. What was the irrigation quantity?

Irrigation was given accordingly to maintain field capacity level in both control and heat treat stress condition to avoid drought stress. Irrigation was given at 3-4 days interval from seedling stage to physiological maturity

28. A blank before non-stress

We have given the space accordingly see highlighted in green

29. Some traits of the soil used in the experiment should be detailed.

We have mentioned the soil used in the pot please see text highlighted in green.

30. Sowing, planting and harvesting dates should be given.

We have mentioned the sowing date and harvesting date please see text highlighted in green.

31. Change “Data analysis” to Data analyses”. There was more than one analysis.

We have changed it accordingly please see text highlighted in green.

32. Duncan or Tukey tests could be used instead of LSD.

We have used Duncan Multiple test range please see Table S2 highlighted in green

Results

33. Remove soil from “Under normal soil conditions,” because NS was explained as non-stress (NS) conditions.

We have removed it accordingly please see in the manuscript highlighted in green.

34. Remove the seed as well. You explained seed analyses before for readers.

We have removed it accordingly please see in the manuscript highlighted in green.

35. Write conditions after NS

We have written it accordingly please see in the manuscript highlighted in green.

36. Change “under high salinity (HS) conditions” to “under heat stress (HS) conditions”

We have corrected it accordingly please see in the manuscript highlighted in green.

37. Check and correct seed yield per plant (SYP) under heat-stress (HS) conditions! 34.9 is higher than NS.

We have corrected it accordingly please see in the manuscript highlighted in green.

Protein and carbon concentration

38. Change “decreased protein concentrations” to “a decrease in protein concentrations”

We have changed it accordingly please see in the manuscript highlighted in green.

39. Change “increased seed C” to “an increase in seed C”

We have changed it accordingly please see in the manuscript highlighted in green.

40. Remove “seed” because you mentioned that analyses were made in the seed in M&M.

We have changed it accordingly please see in the manuscript highlighted in green.

Micronutrients

41. Change “Gockce” to “Gokce”.

We have changed it accordingly please see in the manuscript highlighted in green.

Seed yield per plant

42. Remove “identifying it as a potential candidate for heat tolerance breeding” from here to “Discussion”.

We have changed it accordingly please see in the manuscript highlighted in green.

43. Abbreviations (G, T, HS, NS etc) were repeated. Please insist on full name or abbreviations through text.

We have maintained accordingly please see in the manuscript highlighted in green.

Discussion

44. BR37, a blank before [37].

We have given the space accordingly please see in the manuscript highlighted in green.

45. A blank between to and HS

We have given the space accordingly please see in the manuscript highlighted in green.

46. BR39, a blank before has.

We have given the space accordingly please see in the manuscript highlighted in green.

47. The scientific name of mung bean can be written.

We have written the scientific name of mung bean please see in the manuscript highlighted in green.

48. The common names of Lentil and Lathyrus should be given.

We have written the scientific name of mung bean please see in the manuscript highlighted in green.

49. The following sentence could be written as reference [63] “It is thought that the chickpea genotypes used in this study have different heat stress thresholds, as in Phaseolus species having different heat stress tolerances or sensitivities [63], which affect the variation of some macro and micronutrient contents.” 63. Tene et al., Star of biochemical traits of heat-tolerant and heat-sensitive Phaseolus genotypes in coping with heat stress. Plant Growth Regulation. 2025; doi: https://doi.org/10.1007/s10725-025-01340-4

We have mentioned the reference in the text please see green highlights in the text

Conclusion

50. The following sentence or similar sentence could be added in Conclusion, too. Chickpeas having the higher nutritional values under HS conditions were suggested to use in breeding programs.

References

We have added the sentence accordingly highlighted in green in text.

51. All references should be checked and corrected according to “Guide for Authors”.

We have checked the references and formatted according to journal guideline.

52. The scientific names of plant species should be italicized.

We have italicized the plant species name.

---

## [Decision Letter · Decision Letter 1]

29 Jul 2025

Influence of Elevated Temperature on the Nutritional Profile of Chickpea (Cicer arietinum L.) Seeds

PONE-D-25-29542R1

Dear Dr. Jha,

We’re pleased to inform you that your manuscript has been judged scientifically suitable for publication and will be formally accepted for publication once it meets all outstanding technical requirements.

Kind regards,

Prafull Salvi, PhD

Academic Editor

PLOS ONE

Reviewers' comments:

Reviewer's Responses to Questions

**Comments to the Author**

Reviewer #2: All comments have been addressed

2. Is the manuscript technically sound, and do the data support the conclusions?

Reviewer #2: Yes

3. Has the statistical analysis been performed appropriately and rigorously?

Reviewer #2: Yes

4. Have the authors made all data underlying the findings in their manuscript fully available?

Reviewer #2: Yes

5. Is the manuscript presented in an intelligible fashion and written in standard English?

Reviewer #2: Yes

Reviewer #2: I am satisfied with the corrections on the MN. There may be typing errors in the MN, and you can check them during proof stage of the MN.

**Do you want your identity to be public for this peer review?** For information about this choice, including consent withdrawal, please see our Privacy Policy

Reviewer #2: **Yes: ** Cengiz Toker

---

## [Editor Report · Acceptance letter]

PONE-D-25-29542R1

PLOS ONE

Dear Dr. Jha,

I'm pleased to inform you that your manuscript has been deemed suitable for publication in PLOS ONE. Congratulations! Your manuscript is now being handed over to our production team.

Kind regards,

on behalf of

Dr. Prafull Salvi

Academic Editor

PLOS ONE